# A systematic narrative review of the research evidence of the impact of intersectionality on service engagement and help-seeking across different groups of women, trans women, and non-binary individuals experiencing homelessness and housing exclusion

**Carolin Hess**[1]*, **Zahra Abdulla**[2], **Lydia Finzel**[3], **Antonina Semkina**[1], **Jess Harris**[1], **Annette Boaz**[1], **Jill Manthorpe**[1]

1 Policy Institute, King's College London, London, United Kingdom, 2 Institute of Psychiatry, Psychology & Neuroscience, King's College London, London, United Kingdom, 3 Resource Economics Group, Humboldt-Universität Berlin, Berlin, Germany,

* carolin.hess@kcl.ac.uk

## Abstract

### Background

Women with experience of homelessness face severe health inequalities, with their average age at death being nearly half that of the general population. Recent research emphasises the compound challenges women with homeless experiences face in seeking help, accessing and engaging with support services, but we know little about the influence of different intersectional dimensions on their support access and experiences. The paper aims to review the evidence and critically engage with the impact of gender/sex, race, ethnicity, age, disability, class/poverty, migration status, religion, pregnancy/maternity and sexual orientation on women's homelessness trajectories and engagement with services.

### Methods

We conducted a systematic narrative review of studies in 2023 investigating the impact of intersectional dimensions on women, trans women, and non-binary individuals with experiences of homelessness' engagement with services and help-seeking. Primary qualitative, quantitative or mixed method research, written in English and published after 2010, was included. Narrative methods were utilised in the synthesis and analysis of the research.

### Results

The search identified 4109 articles after deduplication from which 52 were included for review. The findings highlighted intersectional experiences in help-seeking and engagement across housing, healthcare, the police, social services, and voluntary organisations. The women, trans women and non-binary individuals included in the studies reported a

**Data availability statement:** All relevant data are within the manuscript and its Supporting Information files.

**Funding:** This work was supported through doctoral funding from the National Institute for Health and Care Research (NIHR) School for Social Care Research. The funders had no role in study design, data collection and analysis, decision to publish, or preparation of the manuscript.

**Competing interests:** The authors declare no competing interests.

range of barriers, stigma, and discrimination, often rooted in systematic and intersectional disadvantage which delayed or prevented help-seeking and recovery.

## Conclusion

The review investigates the multiple challenges faced by women, trans women and non-binary individuals with experiences of homelessness, highlighting systemic and intersectional disadvantages which impinge on their agency. Changes in policy and practice are recommended to develop more effective person-centred, culturally and gender-sensitive approaches that can transform intersectional dimensions into strengths, empowering women and improving their recovery and engagement with services.

## Introduction

The average age at death for women sleeping rough or in homeless accommodation in England [1] and other high-income countries [2,3] is estimated to be between 41–43 years; nearly half the age of the general population. With increasing evidence that women's homelessness is often hidden [4–7], recent research has focused on understanding women's differential experiences of homelessness compared to men's [8–10], and gender's impact on accessing support and recovery. Women's experiences of domestic and sexual abuse, and verbal violence may impact on their homelessness [9,11–14], with higher rates of anxiety, depression and other forms of mental ill-health than men [15], and forced separation from their children [16]. Trauma and mistrust [17] may negatively impact engagement with services and recovery [10,18,19]. Difficulties in accessing and engaging with preventive or primary care and other healthcare services may result in problems remaining untreated or unresolved, or becoming compounded. A United States of America (US) study, for example, reported that 37% of homeless women have unmet healthcare needs [20]. A systematic review found women with homelessness' experiences of accessing community-based healthcare were poor and adversely impacted their health-seeking behaviours [21]. Further reviews have investigated barriers for accessing to services such as primary healthcare [21], substance use reduction [22], psychosocial support [23], and housing interventions for domestic abuse survivors [24],

Women, of course, are not a homogenous group and other social dimensions can impact their recovery. Access to services is further complicated by additional, often overlapping and intersecting disadvantage [25]. However, these are often addressed separately. For example, systematic and other literature reviews have explored the impact of race and ethnicity on homelessness trajectories and recovery [26], hospice use [27], health and support access for homeless lesbian, gay, bisexual, trans, intersex, queer, questioning and other gender-diverse (LGBTQ+) populations [28] and those living with HIV/AIDS [27]. Further studies have addressed the impact of ageing [29], and gender reassignment on homelessness experiences and interventions [28]. Reviews have also explored interventions among female drug-dependent street sex workers [30], housing interventions for pregnant women who are homeless [31], and when accessing antenatal and/or postnatal healthcare [32]. Despite this, the intersecting disadvantages of women with homeless experiences and their impacts in different support settings are under-reported,

### Aims of this paper

The review addresses the following research question:

How does intersectionality impact women, trans women, and non-binary individuals with experience of homelessness' support trajectories?

As homeless populations experiencing multiple and intersecting disadvantage are often unable to fully access the supports they need [33,34], we sought to investigate how women, trans women, and non-binary individuals engage with different services - namely statutory housing services, temporary accommodation, healthcare, criminal justice (police services, courts), social services and voluntary services; and to what extent these services address or overlook the possibly interacting impact of sex, gender (reassignment), race, ethnicity, age, disability, class/poverty, migration status, religion/belief and sexual orientation. We also aimed to explore how resilience, informal support and stigma impacted how help was sought, offered and received.

Taking an intersectional approach has the potential to inform policy and practice to address the challenges different groups of women, trans women and non-binary individuals with homelessness experiences face when accessing support [35,36].

The protocol is published on PROSPERO (CRD42023440589) for greater transparency, to mitigate research bias and risk of duplication [37]. Due to the inductive nature of the analysis and to improve the quality and focus of the review, not all in the protocol proposed outcomes were considered in this review.

## Conceptual framework

Three key concepts guided the review:

a) **Homelessness**, using the ETHOS definition [38] but specifically including gender-sensitive variations of unsafe, insecure and unstable housing arrangements [39] such as survival sex, refuges, and living with an abusive partner or in crack houses [40] as part of the definition of homelessness. This also included experiences of "domestic abuse and violence" to account for women using domestic abuse and violence services who are not always recognised as homeless [41].

b) **Intersectionality**, a term [42] used to illuminate the overlapping impact of different dimensions of our identities. To operationalise "intersectionality", the review covered eight of the nine protected characteristics outlined in the UK Equality Act 2010. Marriage and civil partnership were not included as a separate characteristic, as the Equality Act only protects these characteristics in employment. The Housing Act 2014 (England and Wales) integrates and extends the UK Equality Act 2010, requiring local authorities to avoid overt discrimination and transphobia in service provision based on various protected characteristics namely sex (interpreted more broadly as 'gender' in this review), race/ethnicity, age, gender reassignment, sexual orientation, religion or belief, pregnancy and maternity, and disability.

c) **support trajectories**, as homeless populations experiencing multiple and intersecting disadvantage require a high level support which they are often unable to access [33,34], support trajectories were analysed by investigating help-seeking and service engagement strategies. The concept of help-seeking is primarily used in health settings, defined as a "complex process where an individual […] seeks out healthcare for a problem or illness" [43, p.281], but was here extended to self-help, social networks and services whose actions impact on people's situations beyond healthcare.

## Materials and methods

Using the methodological guidance for systematic reviews developed by the Joanna Briggs Institute (JBI) [44], research from quantitative, qualitative, and mixed-method studies was integrated together to maximise the usefulness and depth of the findings. The inclusion of

different research designs was to facilitate a more complete understanding of access to support for women experiencing homelessness and insights into the research objectives. The focus on a specific population and experiences of service engagement and help-seeking generated predominantly qualitative evidence.

Acknowledging the diversity within groups of women and identities beyond traditional binary concepts of gender, this review also focusses on the experiences of trans women, and gender-diverse individuals. Given the small number of studies that included gender-diverse identities, we used the term 'non-binary' to summarise findings for identities that do not fall into male or female gender categories regardless of their assigned gender at birth. This includes identities such as two-spirit, agender, genderqueer, gender-fluid and other non-binary gender identities included in the studies.

Using the demographic characteristics of participants as presented in the studies, we recognise that these terms and findings may not fully capture the diversity of (female, trans, and non-binary) identities and experiences.

## Inclusion and exclusion criteria

Primary research data was included, examining the experiences of service engagement and help-seeking for women, trans women, or non-binary populations with experiences of homelessness, explicitly addressing and investigating the impact of intersectionality,

Studies were included if these populations were historically or currently experiencing some form of housing exclusion and seeking help for it, were over 18 years of age, and impacted by other dimensions protected by the UK Equality Act 2010.

Studies were included if they measured, assessed, investigated, or evaluated help-seeking or any type of service engagement and support around their participants' homelessness and had been published after 2010 (including 2010), when the Equality Act came into force.

Papers with mixed-gender samples were included if the analyses and findings highlighted differences in gender and had a sample of more than 50% women or gender-diverse individuals. Similarly, studies focussing on or containing samples of those under the age of 18 years were only included if a large proportion of the sample contained individuals over the age of 18 years or discussed topics regarding pregnancy and maternity which were also relevant to adult women.

Despite the UK focus of this paper and its restrictions to English language materials, we included international evidence of studies in other high-income countries, as defined by the World Bank [45], to capture a broader and more generalisable understanding of intersectionality.

Systematic reviews, literature reviews and book chapters were not included in the analysis, but, if meeting the inclusion criteria, were hand searched for further references and provided background. Editorials, (policy) commentaries, reviews, discussion papers, and conference abstracts were excluded.

Dissertations and grey literature such as organisational reports were searched for mentions of peer-reviewed journal papers and, if available, replaced with the peer-reviewed publication.

## Search strategy

The literature search took place between August 2023 and October 2023. The development of the search strategy was supported by two clinical support librarians, who were involved in the review and piloting of the search strategy.

Keywords and search terms relevant to different contexts were developed after an initial search of systematic reviews on "intersectionality", "women homelessness", "service

engagement" or "help/health-seeking", and in consultation with the two clinical support librarians. This revealed further terms, sometimes historically used, in high-income countries such as precariously housed, vagrant, hobo, street people (US context) [46], sans domicile fixe (SDF) (French context) (ibid.), or houselessness/dwellinglessness, a term increasingly used to emphasise the lack of adequate housing or shelter rather than a lack of a 'home' [47].

Key words with regards to intersectionality were also developed with the help of population filters [48–52].

As there is evidence that low-threshold support services or informal support networks are perceived to be more accessible and attended by particularly marginalised populations [53,54], the search strategy focused on low-threshold services which have minimal or no criteria for access, coping strategies such as self-help initiatives, and reliance on personal networks. Such services are largely community-based, and voluntary or third sector provided, offering outreach programmes, drop-in centres, advice centres, soup runs, emergency services, drug and alcohol or addictions services, homeless shelters and some direct access accommodation. Mainstream low-threshold statutory services in primary healthcare, mental health, housing, and social services, and criminal justice (police services; probation) services which are more likely to be accessed in emergency and crisis situations were also included [34].

The search terms were piloted and adapted by CH after an initial search on MedLine and PsycInfo, limiting the research to high-income countries and English language papers only. Indexation in the MeSH-database was used to identify further terms that were applicable to the concepts. The search terms were also cross-checked by JM.

Following the initial search, the search was transferred to other databases. If available, filters were used, restricting the search to English language papers, and papers published in or after 2010.

## Sources of evidence

Using the search terms identified after the initial search (S1 Appendix), the following databases were searched in August 2023: MedLine, PsycInfo, Applied Social Sciences Index and Abstracts (ASSIA), NHS Evidence Search, OvidSP - Social Policy and Practice, Scopus, Social Services Abstracts, Sociological Abstracts, Web of Science.

Using a limited range of key words containing "women homelessness" and "service engagement" or "health-/help-seeking", British Library e-theses online service (EThOS), Open Grey, Social Care Online, GoogleScholar, WorldCat Dissertations and Theses, and King's College London Library were searched.

Finally, government and organisational reports and reports from sector providers were hand-searched to increase the comprehensiveness of evidence which might not be disseminated otherwise [55] and to keep updated.

Websites of relevance which were searched included Homeless Link, Centre for Housing Policy, Shelter, Revolving Door, St. Mungo's, Crisis, Single Homelessness Project, Fulfilling Lives, Canadian Observatory on Homelessness, Homelessness Australia, European Federation of National Organisations working with the Homeless (FEANTSA), The King's Fund, Joseph Rowntree Foundation, ResearchGate, National Institute for Health and Care Excellence (NICE), and National Institute for Health and Care Research (NIHR).

Weekly electronic alerts were set up to capture new and relevant literature after the initial search was conducted.

All citations identified using the search strategy were imported into the EPPI reference management software for deduplication, facilitate double-screening, data extraction and visualisation [56]. The selection process generated 4,109 initial papers after de-duplication. Abstracts of all of the initial papers were independently screened by CH and ZA, LF, and AS,

following Aromataris and Munn's [44] guidelines. Double-screening reduced the risk of bias in excluding relevant studies [57] and ensured a more objective screening process. A hierarchical screening tool was used [58].

Disagreements were resolved by discussion and reaching consensus. 443 papers were included for full text screening. Three-quarters were double-screened, with consensus reached following discussion.

In total, 48 studies were included which were forward and backward screened for additional sources by CH. Three studies were further included this way.

## Quality assessment

Validity, reliability, applicability of the research design (sampling and data collection) and analysis were assessed using the JBI Appraisal Tools for qualitative and cross-sectional research [59].

Additionally, we assessed the relevance of the studies (the extent to which 'intersectionality' or the intersectional dimension was discussed) to maximise the inclusion of a wide variety of papers. Studies were excluded if they scored below 60% according to the JBI checklist or were deemed of low relevance (n=4) (See S3 Appendix for a summary of the quality appraisal).

A sample of 14 papers (over 20%) was appraised by ZA and LF and discussed with AB and JH. Disagreements were resolved through group discussion.

## Data extraction

Two types of data were extracted by CH between February-April 2024. Firstly, information such as study aims, research design, population (age, homeless status, other demographic variables), year of publication, geographical area and setting, was extracted deductively for analysis.

Secondly, studies were mapped according to the population category, intersectional characteristic(s), mentioned service "needs", and services and help-seeking discussed.

A framework for analysis (S2 Appendix) was developed then piloted for the first five included papers. Subsequent papers were mapped according to this framework, which was discussed in a research meeting with LF and ZA.

## Analysis

Due to the wide range of methods included, narrative methods were utilised in the synthesis of the research to reflect on the qualitative and quantitative evidence [57]. Quantitative data was transformed into themes and included with qualitative data in the narrative analysis.

Following Popay et. al.'s [57] guidance, CH created a preliminary synthesis of tabulation, groupings, and clustering of outcomes. Studies were grouped into subgroups by experiences of intersectionality. These were then tabulated across the different services and support systems accessed when seeking support.

In a second step, relationships among the identified primary themes were iteratively identified, then similarities and differences among the subgroups thematically analysed using NVivo [60] to manage the data.

Analysis of findings, conclusions, and recommendations were discussed by the research team and an advisory group of seven formed as part of a wider study on women's engagement with support and services within CH's doctoral research. The advisory group consists of people with lived experience of homelessness, one practitioner and four researchers with expertise in the field. Four members have lived experience. Their involvement confirmed the appropriateness and relevance of the research findings and increased the authenticity of the findings [61].

## Findings

As shown in Fig 1., 52 studies met the inclusion criteria. They covered qualitative (n=42), quantitative (n=4), and mixed-method (n=6) designs. Most of the included studies were conducted cross-sectionally and lasted under a year.

Studies were predominantly conducted in Anglophone countries, such as the US (n=21), UK (n=9), Canada (n=13), and Australia (n=4). Four studies were conducted in mainland Europe (the Netherlands, Poland, Norway, and the Czech Republic), and one in Israel. The sample size across all papers covered 4,038 participants of different ages, ethnicities, and backgrounds. The majority of these identified as women, approximately 4% (~174) as transgender, and less than 1% (~48) as 'non-binary' and other gender-diverse identities. The studies including transgender and non-binary individuals typically identified participants' gender identities using single-item approaches on surveys and demographic forms. However, one study [62] chose to not specifically ask participants to disclose their gender identity. The exact number of gender diverse identities is unclear as some are overlapping or unspecified. Participants had experienced different forms of homelessness, but most studies (n=32) recruited participants

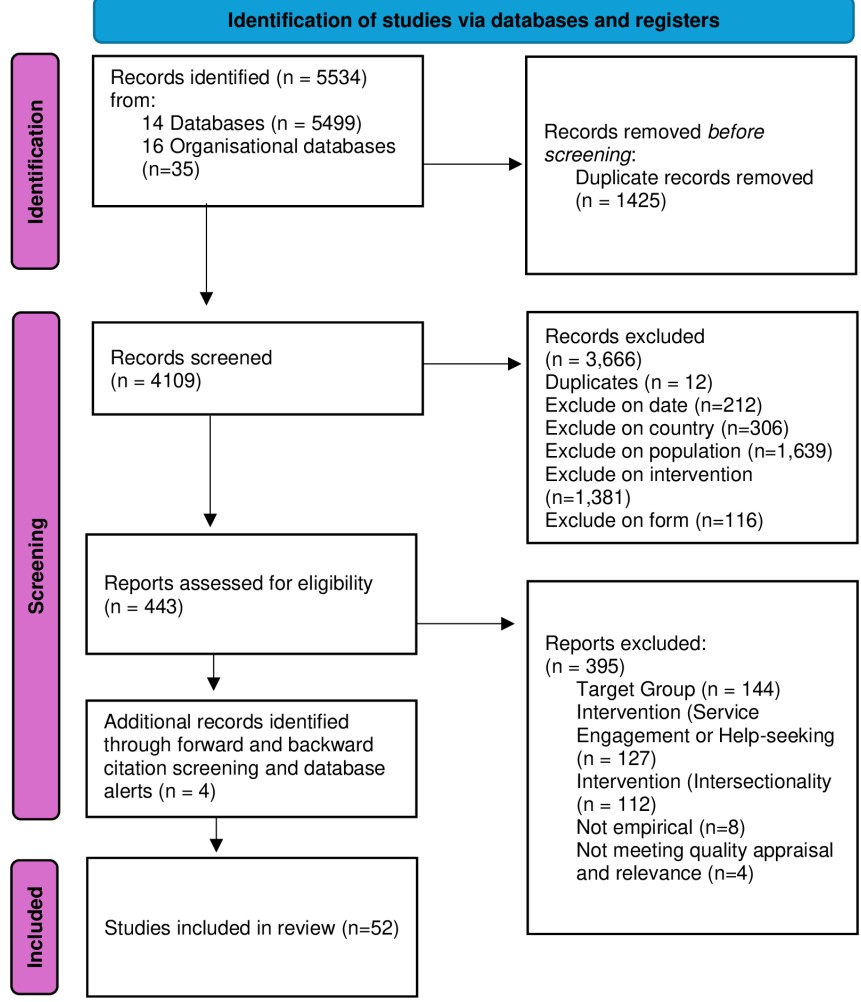

**Fig 1. Prisma Flow Chart.**

from shelters or other temporary and emergency accommodation, including domestic abuse and emergency shelters, specialist women's HIV and LGBTQ+ shelters, as well as mainstream and mixed gender shelters.

Most of the studies (n=48) were journal papers, four were organisational reports. Dissertations and organisational reports were searched for peer-reviewed papers which led to the inclusion of five further papers [14,63–66].

Some papers focused on a particular group of women: the main group (in 18 papers) was pregnant women and (young) mothers, perhaps unsurprising given the gendered dimension of these conditions, four focused on young women, seven on LGBTQ+ and/or specifically trans women (out of which 3 were focusing on LGBTQ+ youth), two on older women, seven on immigrant/migrant populations with no access to public funding (No Recourse to Public Funds (NRPF) in the UK context). The remaining 14 studies did not focus on any subgroup. Families experiencing homelessness are disproportionately headed by single mothers, and many homeless women are mothers, although in most studies, their children were not in their care [14,67–69]. Women who had kept their children and engaged with services were often staying in emergency or temporary accommodation. The impact of this was discussed in six papers [17,70–74].

Race/ethnicity (n=27) was referenced frequently. Ten studies discussed barriers and facilitators for trans women, including three which also discussed the impact of people's sexual orientation. Age was discussed in seven studies, two of which focused on older women's experiences. The small number of papers focusing on "young women" is likely due to our focus on those aged 18+ and does not necessarily point to a research gap. Disability was specifically discussed by few papers (n=4), although this number does not include references to long-term mental health conditions. Poverty/class (n=10), and religion (n=8) were also discussed, but were generally not the primary focus of the studies.

In addition to housing needs, participants predominantly experienced domestic abuse and violence (n=19), mental health needs (n=26), physical health needs (n=10), and substance use (n=16). While we initially intended to focus on low-threshold services (see 'Search Strategy'), engagement with statutory services, such as Social Services (n=11), Housing Authorities (n=20), Healthcare services (n=16), and Criminal Justice systems (n=4) appeared more prominent in the selected studies. However, studies also covered shelters, addiction services, (sexual) health clinics, food banks or employment services. The greater emphasis on statutory services is likely due to the majority of participants being recruited from temporary accommodations and already involved with services.

Limited research covered legal aid and criminal justice in relation to family law and (domestic) violence and abuse, and probation. While we found studies that investigated Emergency Department/ Accident and Emergency (A&E) and hospital utilisation across different groups of women [e.g.,75], we could not find any studies engaging further with the intersectional processes behind these outcomes.

## How intersectionality impacts on women, trans women, and non-binary individuals seeking help and accessing support

Findings are presented in eight themes covering how intersectional dimensions may impact access to housing and healthcare, engagement with social services, voluntary/specialist services and the police, as well as self-help, resilience and stigma.

To highlight the lived experiences of the women, trans women, and non-binary individuals in these studies, to add depth and illustrate the relevance of the findings, some direct quotes from participants in the included studies are provided.

**Priority access to housing.** Poverty and lack of access to affordable, safe and quality housing are major concerns before, during, and after experiences of homelessness

[12,65,74,76–78]. Women from ethnic minority groups, immigrants/refugees and those with no recourse to public funds (NRPF), low-income single mothers [78–81], or who identify as transgender or sexual and gender minority [82], often face additional barriers and discrimination in obtaining housing, and in securing employment necessary to afford private housing.

High-income countries generally have housing subsidies in place for people unable to acquire and pay for housing, such as Section 8 housing choice voucher (US), rent supplement programmes (Canada) or local authority administered housing benefits (UK). They set eligibility criteria and are often resource constrained. Some intersectional factors, such as gender and disability, can improve legal access to statutory housing. In England and Wales, for example, priority is given to homeless applicants who are families with children, pregnant, or vulnerable due to age, disability, or other pertinent reasons under the Housing Act 1996. However, even those with some priority face difficulties in being housed [70,74,83] and multiple and intersecting needs often render these entitlements less accessible. With the shortages of social housing (non-profit rental housing), housing services prioritise access to those in crisis or most at risk, such as women in later stages of pregnancy [83].

Since the Domestic Abuse Act 2021, a UK resident also has priority for (re)housing as a survivor of domestic abuse, but some survivors continue to report not being able to afford alternative accommodation or experiencing long waiting lists. If children are involved, domestic abuse and violence may prompt the involvement of children's protective services, and in some cases, parents may lose the care of their child/ren if they are judged as failing to protect them. In Bimpson et al. [14, p.279], some of the participants' children were removed from their care due to them being exposed to domestic violence in the home. This made one of the women feel as if she were the abusive parent for "letting them [children] see me getting beat up all the time (Nicola)". Further studies have described how the Domestic Abuse Act focuses on physical forms of violence, with emotional or honour-based abuse at times not being recognised, leaving many women, often from minority ethnic backgrounds, without support [84]. Some of the 16 women in Bimpson et al.'s study [14, p.280] felt "punished twice" by their partners and a system which was unable to break their housing dependence which might have helped them to keep their children.

Not having dependent children affects women's priority status for housing [14,79,83]. In the UK context, for example, women not having the care of their children are classed as "single" and only entitled to single person housing, which creates a further problem for women trying to have their children live with them: "I'm only entitled to a one bedroomed flat, […] … how can the children come back to be housed with me if I can't have a big enough house and they won't give you anywhere until the children are back, it's catch 22 (Emma)" [14, p.281]. In Ruttan et al. [85], a mother housed her child with a relative while she slept rough or in hostels, which lowered her priority on housing waiting lists. Without housing she was unable to have her child live with her.

When housing is acquired, it was reported as sometimes being in substandard condition, small sized, in poor neighbourhoods [86], overlooking 'drug corners' [66], cold, mouldy, in need of repair [87], or not safe [71,81,87,88]. Some allocated social housing was at a distance from participants' social networks [89].

**Seeking safety in emergency and temporary accommodation.** While waiting for settled accommodation, or to flee abusive situations, the women, trans women, and non-binary individuals were often offered emergency or temporary accommodation, including shelters, hostels, B&Bs (bed & breakfast) or (domestic abuse) refuges [14,65,71,76,90–96].

Individuals who are temporarily housed, in shelters or refuges, generally report being better connected to doctors, dentists, and other health professionals [88], decreasing anxieties

[65,94,97]. However, the numbers housed in emergency or temporary housing are rising in the UK [98]. Many of those accessing different emergency and temporary accommodation stayed there longer than anticipated [94,99], which negatively impacted their wellbeing [66,100].

As demand for temporary housing increases, groups with additional intersectional disadvantages face further housing disadvantages. Studies reported few women-specific emergency shelters [80], and even fewer shelters specifically for women, trans women, and non-binary individuals with further intersecting dimensions. Specialist services for young women, LGBTQ+ groups, women from ethnic minority groups or with NRPF are particularly rare, and often restricted to urban areas. In the UK, for example, specialist minority ethnic or LGBTQ+ refuges are mostly in London [101,62]. Magill [78] referred to a pre-pandemic report of only one refuge bed per region in England for women with NRPF in 2017, increasing their risks of exploitation and abuse. The COVID-19 pandemic and the increase in demand for refuges, led to even fewer bedspaces for women in general and for women with NRPF. Consequently, it is harder for minoritised women or women with NRPF to find a refuge space [99], and length of stay may also be longer [102]. Shelters may also be unprepared for the healthcare and mobility needs of older and disabled women [103].

Women from gender and sexual minorities and transgender women have reported feeling unsafe, incidences of gender policing, and facing discrimination, such a being placed in male shelters despite identifying as female, or threats, harassment, and psychological violence from staff and other residents [82,62,104,105]. Sex-segregated shelter accommodations can subject trans and non-binary individuals to frequent questioning about their identity, gender expression, and potential rejection [62,106].

Shelters are usually designed as emergency, crisis, or temporary solutions prior to accessing long-term housing [72,87] and participants often reported wanting to move on. With housing shortages, this may take a longer time, as one woman in Benbow et al. [65, p.110] who fled domestic violence, acknowledged: "I left to get away from that, right, I left to get out of the hole, and then just a year later to be in here (shelter), it just makes it feel like none of it is worth it".

**The right to healthcare.** Despite often having high healthcare needs, homeless populations face several barriers in accessing healthcare services. Accessing primary care and mental health services requires time, resources and energy which can generate unique challenges for people in unstable housing who are experiencing multiple intersecting needs [17,74]. Managing severe long-term illnesses and disabilities is particularly challenging for people with limited resources and without a stable place to live [11,66].

Obtaining food, clothing [74,87,97] and basic needs such as phones, and transport [87], as well as essentials like nappies and wipes [74], were sources of worry for many participants. This often takes precedence over healthcare and delays engagement with healthcare services. As a result, they may seek healthcare treatment in emergencies or when in severe pain [11,72]. This can be reinforced by rigid eligibility criteria, often making intensive support only accessible at crisis or in emergencies. A service provider in Benbow et al. [65, p. 183] reported that a mother who needed psychiatric care was unable to acquire the necessary supports "until her illness was so severe that she attempted to die by suicide".

Some young people and transgender individuals experience identity-specific barriers to accessing healthcare services, reporting feeling stigmatised and unable to access (gender-affirming) care [11,101]. From the US, Wagaman [101] described financial and administrative barriers, such as the need for approval from a therapist (which many cannot afford or access) before beginning their medical transition, and lack of resources to afford transition-related expenses to support the transition. Aboriginal women in an Australian study questioned the

acceptability of hospital services, noting that they did not feel culturally safe when attending local public hospitals [76]. Racism being dismissed or pathologized, rather than addressed as a legitimate concern, underscores the systemic biases and discrimination faced by Black, Asian, and minoritised individuals within healthcare systems [90]. Women with NRPF or illegal visa status may be less likely to use healthcare services due to the potential impact on their position [107,108].

Women also generally hold responsibility for their sexual and reproductive health needs. Sexual health clinics can provide safe spaces for homeless women engaged in abusive or exploitative relationships and empower them to negotiate contraception choices and pregnancy [109]. Access to healthcare at early stages of pregnancy can positively impact women's substance use, diet, and the wellbeing of both mother and child. The lack of it increases the risk of complications [88,92,97], perinatal anxiety, depression, and maternal suicide [92]. Yet, some women report accessing help only a few weeks, or even days, before the birth of their baby, leaving them mentally and physically unprepared for motherhood [80]. In the US context [86], pregnant participants have criticised a lack of accommodation and services for pregnant women with partners, which impacted some of their decisions to engage.

Even if able to access prenatal services, support may cease after the birth [110]. Women with a history of drug misuse were reported sometimes to use substances [92,110] to cope with anxiety and depression when postnatal support ceased.

**Social services, poverty and motherhood.** Homeless mothers can experience social services (children's services or child welfare) as punitive and judgemental when not conforming to their expected role of a 'good' mother [63,74,76,86,92,111]. The fear of social service involvement and of losing their children may lead women to conceal their difficulties and not seek treatment [11,110], downplay their substance use [73], or not report abuse [63].

Research has shown that many homeless individuals have been in the care of a local or state authority [68] and felt social services had failed to respond to neglect, sexual and physical abuse during their childhood. Upon turning 18, they felt unsupported and lacking guidance [90]. For some this was compounded by negative experiences during adulthood and experiences of stigma, e.g., when mothers were sex workers [111]. Historical processes, such as the Stolen Generation and removal of children from Australian Aboriginal and Torres Strait Islander families, may lead to reluctance among some communities to engage with social services [76]. As one women noted: "…I still don't want to like put a foot wrong or anything like that…I am afraid even to talk to them, or tell them how I feel, or if I have got a complaint or something." [91, p.14].

Conscious and unconscious bias and stereotypes due to the mother's poverty, substance use, age, sex work, gender identity, and race/ethnicity can lead to parent(s) being perceived as neglectful or abusive [64,74,76,92,111]. Over a third (37%) of sex workers in Duff et al.'s study [111] had a child taken into care. Also, young mothers are more likely to have their babies taken into care than older mothers [63,85]. Rather than feeling that their circumstances were taken into consideration or prompting support, some reported feeling criminalised, scrutinised, and support being unavailable or given too late or only in relation to their pregnancy [77,85]. Women felt solely responsible for their engagement with social services, even if a partner was involved [86], and marginalised as 'maternal outcasts' [14].

When pregnant women felt they were unable to keep the baby or when children were removed this affected the mother's wellbeing. It created further trauma, destabilised their lives, negatively affected their mental health and drug use [12,85,86], and reduced their trust in services [14,96].

These findings have prompted efforts to wrap-around support from teams of general practitioners (GPs) (family physicians), midwives, key workers, and specialist substance misuse

workers [110] to support mothers and the integrity of families. If mothers are adequately supported to keep their children [76,85], pregnancy may represent a "window of opportunity" [83], a "most helpful source" of strength [85, p.40], providing impetus to leave an abusive partner [63], and facilitate behavioural and situational changes that seemed previously too hard [17].

**Criminalising poverty.** Some studies reported negative encounters with the police when women experienced violence from a partner or when staying on the street, particularly due to poverty, race, ethnicity or gender identities. These encounters included being victim-blamed, disrespected, or deterred from reporting further incidences [11,90,104]. Experiences of stigmatisation, for example due to poverty or their sex work, can create mistrust: "The police look down on us. They think, 'Oh you got what you deserved. If you weren't out there it wouldn't happen to you,' you know. But financially, they don't understand the struggles that we have to go through to make ends meet, right?" [11,p.284].

Mistrust of the police also developed because participants were from countries where they felt that the police cannot be trusted [93,107], because they feared deportation [93], police use of excessive force [107], or due to experiences of discrimination based on their gender, sexual or ethnic or racial identities [11,90,104,107]. Particularly gender, racial and ethnic minorities felt disrespected and misunderstood and that police would be delayed, prejudiced, and/or not show up when called [76,78,107]. Langton et. al. [71, p.62] reported slow police responses to calls about abuse and violence in poor areas with a high proportion of ethnic minoritised groups, with assumptions being made about the incidence: "It's like, 'Oh, it's just happening again'". This process has been coined the "criminalisation of poverty" in the context of racialised, transgender sex workers [112] or sex working mothers [111]. These two studies described how sex workers' choices, constrained through poverty and discrimination, are often criminalised and policed, rather than helped through preventive, specialist and gender affirming care and housing.

Henry [113] also highlighted a lack of culturally sensitive responses and a hierarchy of abuse in the Australian criminal justice system, which takes physical abuse more seriously than emotional or online abuse. The latter is less likely to lead to a prosecution of domestic abusers, possibly risking the safety and housing of their (female) partners [14].

However, a trauma-informed culturally sensitive response may help someone access support, as one woman from an Asian background [97, p.41, also in 62] noted: "They understood [my cultural background]; they let me speak in my own time, so it was very positive. Because of this it was much easier [to speak to the solicitor (lawyer)]. Because the police and the solicitor both could speak my language." In a Norwegian context [94], the police often served as contact points, putting women in touch with other services.

**Resilience and informal support.** Despite the multiple disadvantages and challenges women, trans women, and non-binary individuals were facing, many showed resilience and survival skills [11,64,66,104]. Such self-help, but also informal networks, and peer-support can play a crucial role in navigating challenges. However, how and the extent to which they are utilised can also depend on intersectional experiences.

Some of the participants relied on online resources, for example to seek and find out about gender-affirming support, health, and legal information [97,109,113]. For others, prayers and faith [17,70] or having to care or support someone else [66,110] helped in difficult times, enhanced self-esteem, and served as a powerful incentive to stay positive and hopeful [77,89].

Informal support systems were also described as crucial in the help-seeking process when available. Women, in particular, would often exhaust informal options before seeking formal ones [4,41,100]. LGBTQ+ participants in one study who were alienated from their biological family, relied on support from their "gay mothers" [104, p.11250]. Social networks can serve

as a buffer when navigating challenging events, helping with finances, childcare, or social support. In one study, pregnant women who were able to reunite with family were more likely to retain care of their children than those who did not [86].

However, family reunification is sometimes impossible and support networks may be unsafe or dysfunctional [114], manipulative [70], impose further stress [85] and stigmatisation [115]. Some become estranged from families due to abuse or neglect [86]. Family and community may pressurise an individual into certain decisions [76,93] or impose barriers to speak up about matters, such as domestic violence and abuse or substance use [76,78,107]. For some, initial support from friends or family developed into abuse or unhealthy power imbalances [99], strained relationships or led to feelings of burden, compounding families' financial and emotional struggles [70]. Some women, trans women, and non-binary individuals reported not having a reliable social network at all or relying on surrogate families and networks they made on the streets. Experiences of domestic abuse, discrimination, substance use, or migration experiences, for example, may limit social support [11,14,109].

Including peer-support and diverse voices of people with lived experiences in service structures can help efforts to strengthen resilience and build more supportive networks. Peer-support and sharing of experiences with others in similar situations may provide validation, encourage positive decisions and expose people to a wider range of perspectives and intersectional experiences [11,63,71,76,92,97,116,117].

**Stigma and shame.** Stigma or fear of intrusive service involvement is often influenced by intersectional experiences and can become a barrier to seek assistance [11,17,63,66,73,90,110,114]. Feelings of guilt, rejection and not being believed often stemmed from childhood experiences, being reinforced and impacted by uncertain legal rights, traditional gender views and previous discrimination, such as experienced by Black drug users [66], drug user mothers [74], or sex workers [111]. Younger homeless women reported not being taken seriously [90,101]; although these challenges may be reversed for young black women, with professionals attributing greater responsibility and culpability "adultification" for their age [118]. Professionals working with migrants with NRPF in the UK [74, p.9] and the US [107] perceived a certain public's "disdain against them", underpinning the lack of options and hardship they are experiencing.

Attitudes were sometimes reinforced by policy, professionals, the community, family or other parts of the social network. Some professionals interviewed [70,73] blamed women for their choices and circumstances, perceiving them lazy, self-defeating, or oppositional rather than looking beyond their disadvantaged context. A woman in Baumann et al. [11, p.284] had been stabbed during sex work, but decided to not seek medical treatment because she had previously experienced judgmental comments over her clothing. Women reported not wanting to disclose being in abusive relationships [11] or downplaying mental ill-health symptoms or drug misuse during pregnancy [65,74]. Some felt unwilling to discuss childhood trauma and abuse, particularly early in treatment [74]. For women whose cultural or religious backgrounds discouraged open discussion of personal problems, asking for help with certain needs could feel uncomfortable or unfamiliar, "I am supposed to be married forever and not ask for help when things are bad." [104, p.537].

These feelings may mean they remain longer in dangerous and abusive situations [107], and normalise their trauma [17], thus exacerbating reluctance to disclose abuse or seek help.

**Can voluntary and specialist services fill the gap?** Voluntary services are often complementing or filling the gap of statutory mainstream services by advocating and providing services for specific groups. Several of the studies voiced a need for more trans-specific, gender-specific, and/or ethnic-minority specific programmes and support, including emergency and temporary accommodation. Community-based and advocacy

services can provide crucial support and services for marginalised populations falling through mainstream services or feeling "cultural disconnect" [76,99,113,119]. Such services were praised for their responsiveness and providing vital support, resources, and social integration [92].

Voluntary organisations, especially those catering towards a specific community, may serve as pockets of safe spaces while mainstream services are often experienced as inaccessible or exclusive. These spaces of peer and community support may offer some therapeutic care, comfort and practical assistance from people in similar circumstances [12,64,70,73,74,78,81,88,94,101,117].

However, availability and inclusivity of voluntary organisations are dependent on the geographical area, with urban and more affluent areas often seeing a wider range of services (for the UK, see for example [120]). Voluntary services may also not be so bound by anti-discrimination laws, such as the UK Equality Act, Canadian Human Rights Act or equivalent legislation. In one study [105], an indigenous transgender participant was rejected from drug treatment by a Canadian voluntary organisation on religious grounds. In the US [112], a transgender woman was not allowed to wear make-up or women's clothing, so she left the voluntary sector programme. These examples challenge the capacity of voluntary sector services to handle the diverse needs of people.

Also, not every service will cater for the heterogeneity of marginalised populations, particularly where they do not fit dominant group expectations. "Open door policies" in some Canadian hostels promote a more inclusive environment, regardless of people's identity, (dis)ability, ethnicity or needs [116]. But, by assuming a shared identity of homelessness, such spaces often overlook the heterogeneity of identities and varying needs. Trans women and non-binary individuals, in particular, felt unsafe in the space with cis women who might openly criticise their presence, particularly if they would not visibly pass as "women" or be judged "feminine" enough by staff and others [62,105]. There is a risk that some of the inclusive or specialist approaches prioritise dominant within-group experiences and exclude individuals who would not fit normative assumptions of, for example, "motherhood", "race", and "gender".

## Discussion

The review highlights the complex and multifaceted challenges faced by homeless women, trans women, and non-binary individuals based on their identity and status. The study participants faced a range of intersecting disadvantages, including domestic abuse and violence and the effects of coercive control, substance use, housing instability, limited education, poverty, mental and physical pain, involvement with the criminal justice system, and other forms of physical, sexual and verbal violence. These disadvantages are often viewed as outcomes of their individual choices when accessing services. Yet, acknowledging unequal power dynamics and vulnerabilities, their choices about help-seeking and engagement are deeply rooted in systematic, intersectional, and multiple layers of disadvantage. They are embedded in intersectional experiences when being disabled, pregnant, transgender, from an ethnic or racial minority, poor, or without social networks, which cannot be considered in isolation from each other [9,11–14,65,78,107,111].

The findings corroborate previous research findings on women's vulnerability to homelessness due to societal expectations, unequal access to the labour market, economic decline and reduced availability of affordable and accessible housing, (domestic) violence and abuse, and having limited resources and support networks to draw upon [4,71,121–123]. Also, trans women and non-binary individuals experience structural discrimination in employment and housing, and are more vulnerable to family rejection and homelessness [112,124]. Due to the feminisation and criminalisation of poverty, the studies frequently reported that welfare

support and resources were insufficient for participants to address housing, manage long-term illnesses, severe mental, physical and sexual healthcare needs, or other subsistence needs [64,76,78,86,92,99,107,111,112,125]. Homelessness housing, healthcare, and social services are often not designed with diversity in mind, which can make them inappropriate, physically and emotionally unsafe, and discriminating against individuals who do not conform to the 'norm' or majority population. Without having basic needs met and being unable to access appropriate support, some of the women, trans women, and non-binary individuals in the studies engaged in risky behaviour [11], sex work, shoplifting and the drug trade [112,115] to care for themselves and often their partners or families, increasing their vulnerability to further harm and exploitation [77,104].

People's multiple, and often invisible, identities can influence how needs and help-seeking are expressed, e.g., due to cultural differences, historical contexts, rejection of identity and perceived or actual discrimination [126,127].

Stigma and stigmatising perceptions of labels and service access contribute to a potentially high number of women, trans women, and non-binary individuals who delay seeking services to the point of "obligatory emergency intervention" [11,14,65,66,72,79,87,88,93,99,102]. The findings corroborate previous research findings that women may experience more stigma than men in relation to homelessness and drug-use [128–130], as well as additional stigma related to motherhood [131,132] and further intersectional dimensions [133,134]. Women with children, for example, may not disclose or delay disclosing their needs when unstably housed due to concerns that this may trigger the involvement of children's services and the fear of having their child(ren) removed [79,83,85,86]. Participants were also described as managing their behaviour and openness depending on their relationship and level of trust with a service provider [66]. While stigma, identity-based violence, and discrimination persist, there is a need for more safe spaces and identity-affirming specialist support for, e.g., disabled women, people from certain ethnic minoritised backgrounds, or those identifying as LGBTQ+, where they are not required to negotiate their multiple identities and feel physically and emotionally safe.

Intersectional discrimination is also embedded in policies that act as barriers, such as criminal justice policies [14,76], healthcare access linked to someone's visa status [108], rigid housing, disability support and healthcare thresholds [12,66]. Policies based on 'deservingness' and severity of needs; e.g., only in later stages of pregnancy [99], when terminally ill [66], in cases of severe physical violence, or attempted suicide [65] often prevent an earlier way out of the situation. Such policies rarely recognise the impact of discrimination which may lead women, trans women, and non-binary individuals to delay or avoid exposing and verbalising needs and trauma [11,12,66,77,99,101]. Baumann et al. and Meyers et al. [11,128] revealed that women and trans women would not always report injuries and disabilities in quantitative surveys and questionnaires but would talk about them in interviews with researchers, highlighting the difficulty of describing and judging eligibility based on rigid criteria and short assessments.

Intersectional dimensions are overlapping, so services and policy must be flexible enough to respond to the individual's needs, rather than implementing standardised 'one-size-fits-all' approaches. More nuanced approaches that embrace 'difference' and contextuality include active listening, person-centred and trauma-informed care that respects and responds to people's circumstances, identities and culture, can positively impact people's help-seeking and access to support [135]. Ensuring that staff -both frontline and management- are trained and aware of intersectional approaches, such as anti-oppression and culturally/gender-specific training [136,137], can help address stigma, violent and discriminatory behaviours and rebuild trust in services. Support can also be more effective when the workforce of first responders and other professionals, such as police, nurses, community workers, midwives and

emergency responders, are culturally and linguistically diverse and sensitive to culture- and LGBTQ+- specific needs [65,76,87]. Sexual health clinics can be a gateway to vital information, education and care, particularly for young and sex-trading women, trans women, and non-binary individuals [109] where support is offered without judgment, sensitive to histories of sexual violence and to the needs of sexually diverse populations [138]. The inclusion of people with diverse lived experiences and backgrounds is essential in shaping more inclusive and safer services [66,83,117].

Many women in the studies engaged with multiple services (on average 7.3 different services in Schmidt et al. [87] and 5 services in Ben-Porat and Sror-Bondarevsky [102]) which they perceived as disempowering and undermining of control [14,89,94]. Transgender and non-binary individuals may feel even more disconnected from services which can appear unsuitable and exclusive [e.g., 62]. Thus, recent research on homelessness has focused on the importance of continuity of care and overcoming the siloed nature of different support systems which could facilitate engagement and trust of overlooked populations [63,139–142]. In the UK, this has led to efforts to 'integrate care' across voluntary and statutory primary, acute and mental health care initiatives to enhance health service coordination for homeless people [143,144]. England's 2023 reforms for care leavers [145] and the "Ending Rough Sleeping For Good" strategy [146], for example, have committed to more coordinated support for care and prison leavers, hoping to reduce homelessness among populations with elevated risk of homelessness [145].

## Conclusion and limitations

This review aimed to comprehensively examine the different ways in which intersectionality impacts homeless women, trans women, and non-binary individuals' help-seeking and engagement with services. The findings align with other evidence on the gaps in statutory services, mostly in housing, healthcare, social services, and police, and highlight that intersectionality is not always sufficiently addressed, often leading to greater difficulties in finding ways out of homelessness. As statutory and voluntary services remain overstretched and underfunded [147,148], with limited time and resources to actively listen and providing holistic person-centred care, the number of people who fall outside the often-narrow 'deservingness' criteria and the systematic exclusion of certain groups will likely increase.

More research is needed to understand intersectional differences in support access, how and why certain populations are more at risk of exclusion, and how practice and policy may improve. This can help to provide person-centred approaches that respond to the complexity of dimensions and experiences that impact people's decisions and trajectories in and out of homelessness regardless of one's gender, ability, race/ethnicity, age, maternity status and class. When addressing power disparities and paying attention to the multiple identities that people can hold through culturally-and gender sensitive approaches, incorporating lived experiences and diversity into services, several papers commented that it may be possible to turn intersectional dimensions into a strength, empowering and facilitating help-seeking and recovery [12,13,65,74,85,92,107,109,119,149].

Efforts were made to capture the breadth of intersectional experiences across different groups of women, trans women, and non-binary individuals and how systems of inequality create distinct experiences. However, the review also echoes the blurriness of the concept. Many of the papers screened, while examining diverse populations, did not adequately reflect or lacked clarity on how the intersectional dimension was impacting the experiences described. Focusing on the intersection of specific dimensions named in the UK Equality Act 2010 also meant that we had to set boundaries between categories and may have overlooked further dimensions that impact women, trans women, and non-binary individuals.

The decision to exclude studies which do not exclusively focus on gendered experiences may have omitted studies exploring specific intersectional dimensions in more heterogenous populations.

Taking on an international perspective, the review aimed to generate a broader understanding of intersectionality beyond a specific policy context. However, the research was confined to high-income countries and to English-language papers, which may have introduced an inevitable bias towards papers from Anglophone countries, and especially the US given its distinct welfare policy and service landscape. It may also hint at a greater significance of certain intersectional approaches and discussions in the US, notably on discussions on race [150,151]. Included studies show considerable variation in methodology, sample characteristics, research design, and measurement tools, which may limit the generalisability of specific findings. Despite our efforts to include findings relevant to non-binary individuals, the limited sample size and overlapping gender categories mean that conclusions about this group are particularly limited and require further exploration.

Further reviews could investigate the lived experience of homeless women, trans women, and non-binary individuals living with a neurodevelopmental or long-term illness, prison leavers, and those in later stages of recovery. Trajectories out of homelessness with a greater focus on women, trans women, and non-binary individuals' survival skills and resilience, and the role of first responders and relationships with partners, friends, and frontline staff, have not been sufficiently investigated. We also know little about the effectiveness of interventions to engage different groups of women, trans women, and non-binary populations to make service access more inclusive and break intergenerational experiences of homelessness and disadvantage.

## Supporting information

**S1 Appendix. Search strategy**.
(PDF)

**S2 Appendix. Data extraction table (included studies)**.
(PDF)

**S3 Appendix. Quality appraisal**.
(PDF)

**S4 Appendix. Data screening – All studies identified in the literature search (excluded and included studies)**.
(PDF)

**S1 File. PRISMA_2020_checklist**.
(PDF)

## Acknowledgements

Special thanks to the first author's doctoral advisory group for valuable feedback and reflections. We also thank King's College London clinical support librarians for their help developing the search strategy.

## Author contributions

**Conceptualization:** Carolin Hess.

**Formal analysis:** Carolin Hess.

**Funding acquisition:** Jill Manthorpe.

**Investigation:** Carolin Hess, Zahra Abdulla, Lydia Finzel, Antonina Semkina.

**Methodology:** Carolin Hess.

**Project administration:** Carolin Hess.

**Supervision:** Jess Harris, Annette Boaz, Jill Manthorpe.

**Validation:** Zahra Abdulla, Lydia Finzel, Antonina Semkina, Jess Harris, Annette Boaz, Jill Manthorpe.

**Writing – original draft:** Carolin Hess.

**Writing – review & editing:** Carolin Hess, Zahra Abdulla, Lydia Finzel, Antonina Semkina, Jess Harris, Annette Boaz, Jill Manthorpe.

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
