## [Decision Letter · Decision Letter 0]

27 Nov 2024

PONE-D-24-34499A systematic review of the research evidence of the impact of intersectionality on service engagement and recovery across different groups of women, trans women, and non-binary individuals experiencing homelessness and housing exclusionPLOS ONE

Dear Dr. Hess,

Thank you for submitting your manuscript to PLOS ONE. After careful consideration, we feel that it has merit but does not fully meet PLOS ONE’s publication criteria as it currently stands. Therefore, we invite you to submit a revised version of the manuscript that addresses the points raised during the review process.

We look forward to receiving your revised manuscript.

Kind regards,

Lakshminarayana Chekuri, MD, PhD

Academic Editor

PLOS ONE

Journal Requirements:

 This work was supported through doctoral funding from the National Institute for Health and Care Research (NIHR) School for Social Care Research 

This work was supported through doctoral funding from the National Institute for Health and Care Research (NIHR) School for Social Care Research. Special thanks to the first author’s doctoral advisory group for valuable feedback and reflections. We also thank the KCL clinical support librarians for their help developing the search strategy.

 This work was supported through doctoral funding from the National Institute for Health and Care Research (NIHR) School for Social Care Research

4. As required by our policy on Data Availability, please ensure your manuscript or supplementary information includes the following: 

Additional Editor Comments :

Thank you for your scholarly contribution to the topic area of Homelessness. I'd also like to thank the authors for choosing PLOS ONE to publish your findings from this study. Comments from reviewers are provided below. Please review these comments and I suggest address them and resubmit your manuscript. Your timely response would help this study be published and will make it accessible to interested readers across the world. I look forward to reviewing your revised manuscript. I wish you good luck with your future endeavors.

Reviewers' comments:

Reviewer's Responses to Questions

**Comments to the Author**

1. Is the manuscript technically sound, and do the data support the conclusions?

Reviewer #1: Partly

Reviewer #2: Partly

Reviewer #3: Yes

2. Has the statistical analysis been performed appropriately and rigorously? 

Reviewer #1: N/A

Reviewer #2: N/A

Reviewer #3: Yes

3. Have the authors made all data underlying the findings in their manuscript fully available?

Reviewer #1: Yes

Reviewer #2: No

Reviewer #3: Yes

4. Is the manuscript presented in an intelligible fashion and written in standard English?

Reviewer #1: Yes

Reviewer #2: Yes

Reviewer #3: Yes

5. Review Comments to the Author

Reviewer #1: Thanks so much for the opportunity to read this review. I commend the authors on the extensiveness of their search, and their thorough analysis and reporting on this important topic. My general comments have to do with the methodology, namely with the outcomes measured and included settings.

INCONSISTENCY IN OUTCOMES & SETTINGS

Reading through the abstract, introduction, and materials and methods, the following outcomes seem to be mentioned at different times:

1. Impact of intersectionality on homelessness trajectories, recovery and engagement with services (Abstract background)

2. Impact of intersectionality on engagement with services and recovery (Abstract methods)

3. Impact of intersectionality on homelessness trajectories, recovery, and engagement with specific services (Aims of this paper; matches abstract background)

4. How help was sought, offered and received (Aims of this paper – Lines 37-38)

5. How services address intersectionality (Aims of this paper; Lines 41-42)

6. How intersectionality impacts recovery and support trajectories (Aims of this paper; Lines 42-43)

7. Access to support / Experiences of service access and help-seeking (Materials and methods; Lines 71-72)

8. Impact of intersectionality on experience of recovery and help-seeking (Inclusion and exclusion criteria; Lines 82-84)

9. Self-help and service support around homelessness or multiple needs (Inclusion and exclusion criteria; Lines 88-90).

I recognise that there’s overlap between each of these outcomes. However, the nuanced differences between them and the number of times they are expressed differently results in that by the time we begin to read the results, it’s not very clear what will be reported. As well, whereas ‘intersectionality’ is well defined, terms like ‘recovery’ and ‘support trajectories’ are not clear. Does recovery relate to recovery from homelessness? Is recovery from substance use disorder and mental illness included in your definition? This is made further complicated by the fact that themes are not reported to match the research question, but rather emerged through a type of inductive thematic analysis. This is fine – I think – but it would be helpful to more explicitly state how each of these themes tie back into an original, more clearly-defined research questions.

My understanding of the review question becomes even more convoluted when I look at the SR’s pre-registration where the main outcomes are again different to what is reported in the manuscript.

I also noticed some inconsistencies with the target setting. In Lines 43-46, the authors mention, “Refuges and day centres, community mental health support, primary care, social care, drug and alcohol services, local authority housing services and other statutory and voluntary agencies, and self-help initiatives.”

There is no emphasis on low-threshold criteria here, as there is further on in the text (Lines 123-129). As well, the criminal justice system comes up in the results but was not mentioned in the methods as part of the included services/settings.

Does criminal justice fall under the “other statutory and voluntary agencies” mentioned in Lines 43-46?

In sum, where the target population is clear, the target outcomes and included interventions (i.e., support services in question) are not which detracts from the meaning we can glean from results and their implications.

OTHER COMMENTS

If possible, I would recommend stating both race and ethnicity (rather than just race) in the target population to ensure that Ethnic groups who experience high rates of homelessness such as Travellers are not excluded.

Typo line 41: The review address

Lines 239 – 242: Who is part of this advisory group? This seems to have popped out of nowhere!

Line 253: Worded as if the (n=32) refers to individual women rather than included studies.

Line 276-277: I’m a bit confused by the sentence, “Poverty/class and religion were discussed only in relation to other variables when seeking help and support.”

Type line 367 – Solution(s) not solution

Typo line 376 – and delay(s)

The quote on Lines 409-411 lacks some context. Is the participant saying that they didn’t get to do the nice fluffy stuff? Is the participant homeless herself or a healthcare provider speaking about the situation?

The review set out to assess the impact of different intersectional dimensions on women’s, trans women’s, and non-binary individuals’ with – potentially – additional protected characteristics on women’s homelessness trajectories, recovery, and engagement with services. Perhaps in part due to some of the methodological confusions I have outlined, I still feel that there is a lack of clarity of how specific identities contribute to these specific elements (e.g., homelessness trajectory, recovery, service engagement and access). I think this murkiness can be resolved by (1) more clearly defining the research question and objectives, (2) more clearly demonstrating how each section of the results answers that research question, and (3) by adding implications/recommendations relevant to specific intersectional identity groups based on your review findings (e.g., how, specifically, can we better support women, trans women, and non-binary individuals of racial and ethnic minority status, of particular ages, sexual orientations, who are pregnant, and/or who have a disability).

Reviewer #2: Greetings esteemed authors. On one hand, I applaud any attempt to increase the visibility of transgender people and non-binary people, and I feel the manuscript I reviewed contains numerous attempts to do so, so I applaud that effort as a transgender person. On the other hand, I don't think the inclusion of nonbinary people is justified by the research corpus you examined. There are many points in the paper where you invoke the population of interest -- "women, transgender women and nonbinary people" -- where none of the cited research comes from the ~5 articles that explicitly mention nonbinary. It is not appropriate to point to a collection of research articles that make no mention of nonbinary people and imply that it has something to say about nonbinary people. Every place where nonbinary people are invoked in this draft, you need to scrutinize the specific articles you are referring to, and remove "nonbinary" from your description of any literature that doesn't explicitly include nonbinary participants.

Of course, there are a handful of subsections where you do cite the specific nonbinary-inclusive citations -- in those cases, it is justified to invoke that group, "women, transgender women and nonbinary people". However, I do think you need to provide some summary statistics about your literature pool -- how many articles include nonbinary people, how many total nonbinary people were considered in each study, or other indications to validate that your conclusions are being fairly ascribed to a group including nonbinary people.

I also think you've missed a very basic and important step in any research involving sexual orientation and gender identity (SOGI) -- you've not provided any operational definition of what "woman, transgender woman or nonbinary" means. Even after reading, I am not certain if you mean "nonbinary people who were assigned female at birth", "nonbinary people who were assigned male at birth", or both. To justify invoking nonbinary people as part of the population of interest explored in these papers, you need to be specific and consistent in your terminology usage, and you need to consider and discuss how the papers under consideration measured or judged gender.

Some evaluation of the quality of SOGI research methods used in each article would likely shrink your literature pool, as it's unfortunately too common for people to invoke nonbinary people in describing a sample, even when their data largely describe other groups.

Reviewer #3: The study aims to understand and summarize existing literature exploring the intersectionality of gender and race and other critical demographic variables on women, trans women, and non-binary individuals’ experience of homelessness, recovery, and engagement with specific services. The authors included research from qualitative, quantitative, and mixed-methods research to provide a more holistic view of the findings. Methods are robust, thorough, and appropriate for this study design. 52 studies from all over the world were included in the final review. They did a great job outlining results in extended detail. It was interesting to learn of the views toward voluntary services versus mainstream services (more exclusive and inaccessible). Findings are corroborated by previous research. Glaringly, intersectionality is not sufficiently addressed in most research, and most support groups also favor persons who fit normative assumptions of a certain group of people versus marginalized. The paper adds to the existing literature around the experience of homelessness and is befitting of publication without revision.

6. PLOS authors have the option to publish the peer review history of their article (what does this mean? ). If published, this will include your full peer review and any attached files.

**Do you want your identity to be public for this peer review?** For information about this choice, including consent withdrawal, please see our Privacy Policy .

Reviewer #1: No

Reviewer #2: **Yes: ** Evelyn Jolene Olansky

Reviewer #3: No

---

## [Author Response · Author response to Decision Letter 1]

13 Jan 2025

Manuscript PONE-D-24-34499

Response to Reviewers and Editor

Date: 11/01/2025

Dear Dr. Chekuri,

We thank you and the reviewers for the thorough feedback on our systematic review “A systematic review of the research evidence of the impact of intersectionality on service engagement and help-seeking across different groups of women, trans women, and non-binary individuals experiencing homelessness and housing exclusion” and giving us the opportunity to submit a revised draft.

We appreciate the time and efforts dedicated to providing feedback to our paper which have greatly improved the quality of our manuscript. We hope we have adequately responded to the comments and addressed all concerns. Please see below for a point-by-point response to the editor’s and reviewers’ comments. Changes are highlighted in yellow within the manuscript.

Thank you again for your careful evaluation and we look forward to hearing from you.

Editor

PLOS ONE style: We have ensured that the manuscript fitted PLOS ONE style, adjusting the title page, headers, and removing the funder from the acknowledgement section.

Financial Disclosure: Please add the financial disclose to our submission form “The funders had no role in study design, data collection and analysis, decision to publish, or preparation of the manuscript." References to the funder have been taken out of the acknowledgement section.

Table identifying all screened studies: We added a numbered table of all studies identified in the literature search, including those excluded from the analyses and the reasons for exclusion in the supplementary material. The table was extracted from EPPI, including first author of the screened study, title, and year of publication. If required, we can also share the exported files from the different databases, which would make it easier to replicate the screening process. Unpublished studies that we were unable to access were excluded and appear as “Excluded on form” in the Prisma Flow Chart (p.10/11).

Data extraction table: The table with the primary data extracted is included in S2 Appendix. The first author (CH) extracted the data between Feb-April 2024 (line 276). Findings of included studies were imported to NVivo and thematically analysed (as noted on line 286). A copy of the themes analysed on NVivo can also be provided if required.

Missing Data: Data was qualitatively analysed using narrative methods. As we did not conduct any statistical analysis, there was no concern over missing data.

Risk of bias/certainty assessments: No risk of bias and quality/certainty assessments was conducted.

Captions: We have included captions on the Supporting Information files at the end of our manuscript, and updated any in-text citations to match accordingly.

Reviewer #1

Target outcomes and interventions are not clear: We have reformulated the Research Question (RQ) (lines 78-79) to emphasise the review’s focus on the impact of intersectionality on service access and engagement. We have also included a brief explanation regarding the discrepancy between the review and the protocol (lines 93-95).

We have taken out references to “recovery” from title, RQ and methods, as we agree with the reviewers that analysed papers had not sufficiently addressed recovery (in terms of drug and alcohol dependency). We have also added a definition of support-trajectory, service engagement and help-seeking (lines 114-120). Finally, we have made efforts to more clearly demonstrate how the results address the RQ, by a) clearer transitions to the findings when appropriate (e.g. 553-555) and b) restructuring the discussion to more directly respond to the original question (e.g. lines 699-707).

Support services in question (i.e. focus on low-threshold services) are not clear: We also provided an explanation for the lack of emphasis on low-threshold services (lines 344-350). Contrarily to what was expected and what we aimed for in the search strategy (line 178), included studies had a greater focus on statutory services.

Other inconsistencies addressed: We have also addressed the other inconsistencies with target setting, and typos. We mentioned criminal justice more specifically as a support service in question (line 83), and added ethnicity to ‘race’ (e.g. line 85). We also gave more detail about the advisory group (line 297) and removed the quote on line 465.

Recommendations on specific intersectional identity groups based on review findings: We felt that including specific recommendations for specific groups would not necessarily reflect our findings and the concept of intersectionality. The different dimensions were shown to be closely intertwined and will need to be examined in relation to each other rather than separately. However, we have emphasised this dilemma more clearly in the discussion and have given some recommendations on how intersectionality in general, inequalities and power dynamics could be addressed (e.g. lines 684-687; 699-704).

Reviewer #2

Justification of including ‘non-binary’: Despite only a small proportion of included studies (and participants within them), where participants self-identified as ‘non-binary’, we chose not to remove references to this group. We felt that doing so may even further contribute to their invisibility and exclusion. Instead, we emphasised (line 310) that non-binary individuals made up of a very small proportion of participants in the included studies and limitation of this (line 767). We also provided participant summary statistics on gender (women, transgender, and non-binary identities) and a brief explanation on how gender was assessed and evaluated in the studies (lines 312-315). We also removed references to non-binary if they were not specifically invoked in the included studies, and sometimes replaced the construct of ‘women, trans women, and non-binary’ with ‘participants’/’people’ when we felt that the theme could more generally apply to the homelessness population. In a few places, we have added in barriers that were mentioned more explicitly to non-binary and gender-diverse individuals in the papers (e.g. line 434).

operational definition of gender: We have provided a better operational definition of how gender was defined (lines 131-135).

---

## [Decision Letter · Decision Letter 1]

5 Feb 2025

PONE-D-24-34499R1A systematic review of the research evidence of the impact of intersectionality on service engagement across different groups of women, trans women, and non-binary individuals experiencing homelessness and housing exclusionPLOS ONE

Dear Dr. Hess,

Thank you for submitting your manuscript to PLOS ONE. After careful consideration, we feel that it has merit but does not fully meet PLOS ONE’s publication criteria as it currently stands. Therefore, we invite you to submit a revised version of the manuscript that addresses the points raised during the review process.

We look forward to receiving your revised manuscript.

Kind regards,

Lakshminarayana Chekuri, MD, PhD

Academic Editor

PLOS ONE

Journal Requirements:

Additional Editor Comments:

Please review attached document.

Reviewers' comments:

Reviewer's Responses to Questions

**Comments to the Author**

1. If the authors have adequately addressed your comments raised in a previous round of review and you feel that this manuscript is now acceptable for publication, you may indicate that here to bypass the “Comments to the Author” section, enter your conflict of interest statement in the “Confidential to Editor” section, and submit your "Accept" recommendation.

Reviewer #1: (No Response)

Reviewer #2: All comments have been addressed

2. Is the manuscript technically sound, and do the data support the conclusions?

Reviewer #1: Yes

Reviewer #2: Yes

3. Has the statistical analysis been performed appropriately and rigorously? 

Reviewer #1: N/A

Reviewer #2: I Don't Know

4. Have the authors made all data underlying the findings in their manuscript fully available?

Reviewer #1: Yes

Reviewer #2: Yes

5. Is the manuscript presented in an intelligible fashion and written in standard English?

Reviewer #1: Yes

Reviewer #2: Yes

6. Review Comments to the Author

Reviewer #1: My main concerns during the first review related to unclear study aims / objectives and how this impacted the reader's ability to interpret the results. This revised version of the paper resolves those issues and, as a result, is much easier to read and understand. With the exception of a typo in the second line of the Abstract ('...half that of that'), I have no further requests for revisions. Well done on strengthening the paper as you have.

Reviewer #2: Thank you to the authors for providing more detail on the proportion of considered sources specific to transgender women and nonbinary individuals. I strongly agree with the authors that the visibility of trans and nonbinary people in research is important and worth protecting -- that said, if we allow such research to proliferate without requiring detailed substantiation and contextualization of the claims therein (chiefly that those claims stem from someone somewhere demonstrably talking to a queer person), then I do not believe these or any populations necessarily benefit. Thank you for further substantiating the extent to which your research corpus considered gender diversity.

7. PLOS authors have the option to publish the peer review history of their article (what does this mean? ). If published, this will include your full peer review and any attached files.

**Do you want your identity to be public for this peer review?** For information about this choice, including consent withdrawal, please see our Privacy Policy .

Reviewer #1: No

Reviewer #2: **Yes: ** Evelyn Jolene Olansky

---

## [Author Response · Author response to Decision Letter 2]

5 Feb 2025

Dear Dr. Chekuri,

We thank you and the reviewers for your response on the revision of our systematic review “A systematic review of the research evidence of the impact of intersectionality on service engagement and help-seeking across different groups of women, trans women, and non-binary individuals experiencing homelessness and housing exclusion” and giving us the opportunity to submit a revised draft.

Please see below for a point-by-point response to the editor’s comments. Changes are highlighted in yellow within the manuscript. We hope we have adequately responded to the comments and addressed all concerns.

Thank you again for your careful evaluation and we look forward to hearing from you.

Title: Please consider revising the phrase "A systematic review..." to "A systematic narrative review...."

- We changed all mentions of “systematic review” to “systematic narrative review”

Line 17 and Line 46: Please consider revising the phrase "age of death" to "age at death"

- We changed “age of death” to “age at death” at line 17 and line 46.

Line 17: Like other reviewer pointed out please delete the duplicate phrase "that of" from this Line.

- We deleted the duplicate.

Line 47: Please consider revising the word "contexts" to "countries"

- We replaced “contexts” with“countries”

Line 60: Please consider revising the phrase: "barriers to access to services" to "barriers for accessing services"

- We rephrased the phrase.

Line 68: Please consider deleting comma: "queer, questioning"

- Instead of deleting the comma, we deleted the “and” before “queer, questioning”

Page 10-11: Fig: 1: The math is not adding up. Please proofread the Prisma Flow chart and update the manuscript accordingly. For instance, after adding 14+16+5534-1425+4, the result is 4143 and not 4109 as depicted in the figure. Please explain.

- Thank you for pointing this out. The numbers added up, but the way we had noted them down was indeed confusing. We have changed the way we have listed the different databases (which the 14 and 16 had referred to) and the numbers of studies we have screened (4109). The additional 4 studies were added after the first screening process (line 214/227) and thus not included in the 4109.

---

## [Editor Report · Decision Letter 2]

27 Feb 2025

PONE-D-24-34499R2A systematic narrative review of the research evidence of the impact of intersectionality on service engagement across different groups of women, trans women, and non-binary individuals experiencing homelessness and housing exclusionPLOS ONE

Dear Dr. Hess,

Thank you for submitting your manuscript to PLOS ONE. After careful consideration, we feel that it has merit but does not fully meet PLOS ONE’s publication criteria as it currently stands. Therefore, we invite you to submit a revised version of the manuscript that addresses the points raised during the review process.

Please review attached document.

We look forward to receiving your revised manuscript.

Kind regards,

Lakshminarayana Chekuri, MD, PhD

Academic Editor

PLOS ONE

Journal Requirements:

**Additional Editor Comments:**

Please review attached document.

---

## [Author Response · Author response to Decision Letter 3]

2 Mar 2025

Line 241 (Prisma): We thank you for pointing this out and have changed the number to “3663”

Line 250 (Prisma): We have changed the number to 395

Line 298 (Please describe how many people were part of the advisory group and a break up of number of people with lived experience of homelessness and number of researchers): We have added the number of people who are part of the advisory group. However, there is overlap between “people with lived experience” and practitioners/researchers. One of the researchers and one of the practitioners also have lived experience of homelessness.

We added a sentence that describes how many people are practitioners/researchers and how many of the advisory group have lived experience.

Line 390 (Quotation): We brough (Nicola) into the quotation.

Line 633 (Identify): We corrected the typo.

---

## [Editor Report · Decision Letter 3]

4 Mar 2025

A systematic narrative review of the research evidence of the impact of intersectionality on service engagement and help-seeking across different groups of women, trans women, and non-binary individuals experiencing homelessness and housing exclusion

PONE-D-24-34499R3

Dear Dr. Hess,

We’re pleased to inform you that your manuscript has been judged scientifically suitable for publication and will be formally accepted for publication once it meets all outstanding technical requirements.

Kind regards,

Lakshminarayana Chekuri, MD, PhD

Academic Editor

PLOS ONE
---

## [Editor Report · Acceptance letter]

PONE-D-24-34499R3

PLOS ONE

Dear Dr. Hess,

I'm pleased to inform you that your manuscript has been deemed suitable for publication in PLOS ONE. Congratulations! Your manuscript is now being handed over to our production team.

Kind regards,

on behalf of

Dr. Lakshminarayana Chekuri

Academic Editor

PLOS ONE